# Efficient DNA fluorescence labeling via base excision trapping

Yong Woong Jun[1], Emily M. Harcourt [2], Lu Xiao[1], David L. Wilson[1] & Eric T. Kool [1] ✉

Fluorescence labeling of DNAs is broadly useful, but methods for labeling are expensive and labor-intensive. Here we describe a general method for fluorescence labeling of oligonucleotides readily and cost-efficiently via base excision trapping (BETr), employing deaminated DNA bases to mark label positions, which are excised by base excision repair enzymes generating AP sites. Specially designed aminooxy-substituted rotor dyes trap the AP sites, yielding high emission intensities. BETr is orthogonal to DNA synthesis by polymerases, enabling multi-uracil incorporation into an amplicon and in situ BETr labeling without washing. BETr also enables labeling of dsDNA such as genomic DNA at a high labeling density in a single tube by use of nick translation. Use of two different deaminated bases facilitates two-color site-specific labeling. Use of a multi-labeled DNA construct as a bright fluorescence tag is demonstrated through the conjugation to an antibody for imaging proteins. Finally, double-strand selectivity of a repair enzyme is harnessed in sensitive reporting on the presence of a target DNA or RNA in a mixture with isothermal turnover and single nucleotide specificity. Overall, the results document a convenient and versatile method for general fluorescence labeling of DNAs.

Fluorescence-labeled DNA is a fundamental workhorse for analysis in basic and applied biosciences. DNA labeling is used in many detection and amplification methodologies, and is used both in short synthetic DNA probes and in long polymerase-synthesized DNAs[1]. Fluorescence offers high sensitivity, with single-molecule detection possible for brightly labeled analytes, and many wavelength options confer broad utility. While non-covalent dyes that associate with DNAs have proven useful for in vitro applications such as real-time PCR, covalent attachment enables greater specificity by localizing fluorescence to one species[1]. In addition, covalent labeling offers multiplexing capabilities as well as in vivo imaging[2]. Single dye labels are useful in primers and small fluorescent DNA probes, and multiple labels in the same strand enhance detection in longer DNAs for applications in amplification and single-molecule imaging via fluorescence in situ hybridization (FISH)[3]. Beyond basic and applied biosciences, the use of multiple labels in DNA is also under study for applications in self-assembled DNA nanostructures and in molecular optics[4].

As a result of this broad utility, many methods for covalent labeling of DNA have been developed[1, 5], and are classified broadly into strategies for direct incorporation of fluorophores into DNA during chemical or enzymatic DNA synthesis (co-synthetic labeling), or post-synthetic DNA labeling by forming bonds with reactive groups (such as amine groups) that have been incorporated beforehand. Direct incorporation during chemical synthesis of oligodeoxynucleotides (ODNs) with fluorophore-conjugated phosphoramidite reagents offers high yields and precise positioning, but these are notably unstable and costly compounds, and their use is limited to DNAs shorter than ~200 nt and to laboratories with oligonucleotide synthesis capabilities. For longer DNAs, fluorophores can be directly incorporated using fluorescent nucleoside triphosphates during the synthesis. However, their incorporation during polymerase synthesis imposes limits on what structures will be well accepted in the enzyme active site, and also presents restrictions on where dyes can be incorporated[6].

[1]Department of Chemistry, Sarafan ChEM-H Institute, and Stanford Cancer Institute, Stanford University, Stanford, CA 94305, USA. [2]Department of Chemistry, Le Moyne College, Syracuse, NY 13214, USA. ✉e-mail: kool@stanford.edu

Post-synthetic labeling of fluorophores can bypass the limitations of enzyme-substrate tolerance by allowing one to use modified nucleotides containing smaller reactive handles (such as azide and alkyne) as substrates for polymerases[1]. Additional promising strategies for post-synthetic labeling involve the use of DNA methyltransferases and β-glucosyltransferase with modified S-adenosyl methionine cofactors as a fluorophore donors[5]. These enzymatic labeling methods have been successful for labeling of nucleic acids at the post-synthetic stage in a site-specific manner in vitro. However, the attachment of multiple dyes via such methods relies on achieving very high yields of bond formation between the dyes and the reactive handles introduced. In addition, the post-synthetic labeling method typically requires multiple washing steps to remove residual cofactors and the large excess of unreacted dyes. While this is widely done in cellular analysis of DNA synthesis via fixation and intensive washing steps[1], such strategies are less applicable where washing is not as readily achieved (see comparisons in Supplementary Table 1).

Ideally, methods for DNA labeling will yield bright signals for sensitive detection and imaging. Unfortunately, many dyes are quenched by DNA bases, lowering signal[7]. While one can in principle add multiple labels in a sequence to enhance emission, this can be also be limited by self-quenching; for example, the emission of an ODN with five fluorescein labels was less bright than one with a single label[8]. Thus, there remains a need for fluorescence labeling that is rapid, inexpensive, and yields bright signals in DNAs of any length. Ideally, methods could be applied without specialized chemical expertise using commercially available reagents.

Here, we report an in situ DNA labeling strategy for oligonucleotides as well as dsDNA that makes use of aldehyde-reactive rotor dyes, also known as universal base excision reporters (UBER)[9, 10], to trap AP sites resulting from excision of deaminated DNA bases. This methodology includes (i) chemical or enzymatic synthesis of a DNA strand containing one or more deaminated bases, (ii) base excision repair (BER) in which a bacterial glycosylase excises the deaminated base(s), generating reactive AP sites, and (iii) oxime bond formation between the AP site and light-up UBER dyes (Fig. 1a). In other post-synthetic labeling methods utilizing reactive handles, the synthesized DNAs require separation from the precursors before labeling as both modified DNAs and precursors are reactive. However, in this work, the precursors of deaminated bases such as dUTP and dITP are not substrates of glycosylases until they are incorporated into DNAs, which obviates a separation step. Moreover, ultrafast oxime formation by UBERs with the AP sites affords light-up signals (up to 500-fold)[9] providing a rapid and wash-free labeling process. Also importantly, these reactions are orthogonal to each other, enabling in situ DNA synthesis, repair, and labeling in one tube, by simply combining all the commercially available components without any chemistry setup, rendering the procedure remarkably simple and approachable to a broad range of scientists.

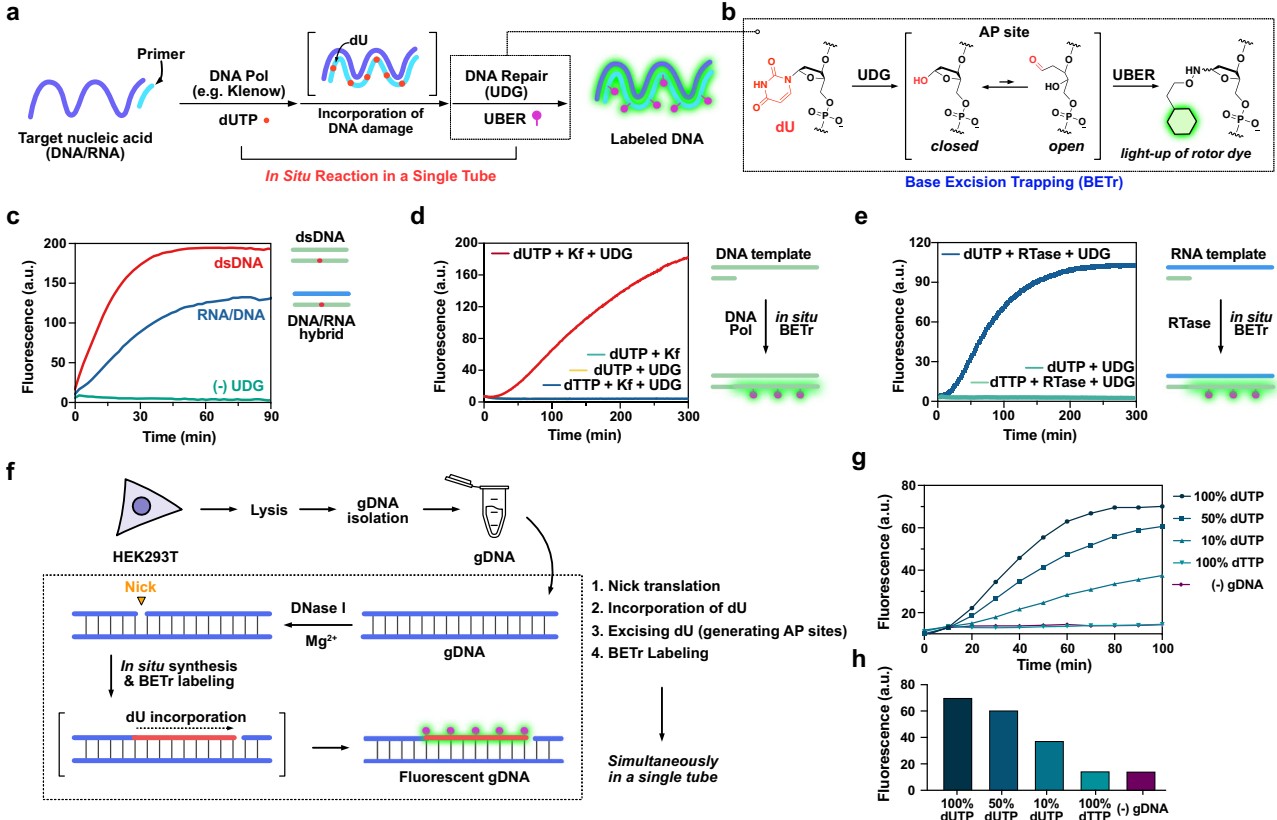

**Fig. 1 | In situ DNA synthesis and BETr labeling. a** Illustration of the incorporation of deaminated bases during DNA synthesis followed by DNA repair and labeling by UBER. **b** Mechanism of Base Excision Trapping (BETr). **c** Fluorescence intensity changes upon BETr labeling in dsDNA (DNA_dU/DNA, 400 nM) and DNA/RNA hybrid (DNA_dU/RNA, 400 nM). **d** Fluorescence intensity changes of in situ DNA synthesis and BETr process with 2 U/mL Klenow (exo-). **e** Fluorescence intensity changes of in situ reverse transcription and BETr labeling with 2 U/μL Revert Aid H Minus RT. **f** Illustration of in situ BETr labeling of gDNA through nick translation. **g** Fluorescence enhancement during the nick translation and in situ BETr labeling depending on the relative concentration of dUTP over dTTP. **h** Fluorescence intensity comparison depending on the relative concentration of dUTP after 100 min of in situ labeling. [CCVJ-1] = 20 μM, [UDG] = 10 U/mL (**c–e**), 20 U/mL (**g, h**), [Target ODN] = 1 μM (9dA), 0.5 μM (RNA), [gDNA] = 40 μg/mL, [Primer] = 5 μM (Primer), 1 μM (RT primer), [dNTP (A,C,G)] = 50 μM, [dUTP] = 100 μM (**c–e**). [dUTP+dTTP] = 100 μM (**g, h**). The fluorescence intensities were measured on **c, d, g, h** a Fluoroskan Ascent microplate reader (485 nm/538 nm) or **e** a Bio-Tek Synergy HT microplate reader at 37 °C.

## Results

### Labeling strategy

A previous study described the synthesis of rotor dyes conjugated to aminooxy groups (UBER) and their application in assays for DNA base excision repair activity[9]. The best-performing probe was CCVJ-1, which contains a cyanovinyl julolidine dye and a three-atom linker to an aminooxy group (Supplementary Fig. 1). Two unusual features of the reaction of CCVJ-1 with AP sites in DNA are (a) an extremely rapid reaction rate, with rate constants up to ~400 $M^{-1}$ $sec^{-1}$ at neutral pH, and (b) a high degree of light-up of the rotor dye, which is dark in solution but gives up to 500-fold enhancement upon covalent binding with AP sites in DNA. These features enabled its use in imaging DNA repair in cells and tissues; it was found that the dye in DNA is highly photostable and can be observed with common fluorescein filter sets, and can also be imaged by two-photon excitation[10].

Given these potentially useful properties, we envisioned applying CCVJ-1 and related UBER probes as general fluorescent labels for oligodeoxynucleotides as well as dsDNA. The protocol would involve replacing deaminated DNA bases with such dyes in a single reaction, using inexpensive bacterial repair enzymes to excise deaminated bases (e.g., uracil and inosine) placed there during synthesis (Fig. 1b). We refer to this as Base Excision Trapping (BETr). While earlier studies have described the incorporation of fluorescent labels into AP sites for detecting products of DNA repair[11–13], previous dyes were fluorescence quenched by DNA, giving low signal intensity and no light-up response[8]. For the DNA synthesis in the BETr method, we envisioned incorporating deaminated bases at one or more sites, either chemically via commercially available phosphoramidites, or enzymatically with DNA polymerase and available deaminated nucleoside triphosphates. Chemical DNA synthesis enables complete control over the location of deaminated bases, and can be performed commercially at reasonable cost unlike the chemical synthesis of DNA with fluorophores. On the other hand, polymerase-mediated synthesis could potentially enable the construction of long DNA strands containing many labeling sites, and might be performed in biological mixtures using target-specific primers. Finally, a DNA polymerase reaction, the enzymatic base excision, and the aminooxy labeling could potentially be performed in a single reaction, making the labeling process rapid and simple. Thus, we performed a series of experiments to test these various labeling possibilities.

### Uracil-containing DNA and in situ covalent labeling

We examined the time required for BETr labeling by bacterial uracil-DNA glycosylase (UDG) excising deoxyuridine (dU) in DNA, and employing CCVJ-1 to form a covalent dye link with a dsDNA containing one dU (dsDNA_dU, Fig. 1c). The simultaneous excision and labeling process was complete in ca. 30 min, and fluorescence enhancement was highly reproducible and proportional to the amount of DNA present (Supplementary Fig. 2). Further experiments with an RNA complement showed that UDG excises a uracil base from both dsDNA and a DNA/RNA hybrid, and CCVJ-1 traps AP sites in both structures with fluorescence enhancement. This efficiency was somewhat surprising for the DNA/RNA case, as the native substrate of UDG is thought to be dsDNA[14]. The labeled DNAs could be isolated conveniently either by size exclusion spin column or by precipitation.

MALDI-TOF mass spectrometric analysis showed that UDG excised the deaminated base almost quantitatively (Supplementary Fig. 3). Notably, the UBER link stabilized the abasic site in DNA; in the absence of the dye, polyacrylamide gel (PAGE) gel analysis showed that 40–50% of the excised DNA was fragmented, likely through known elimination processes (Supplementary Fig. 4)[14]. However, in the presence of UBER to trap the AP site, the DNA was labeled almost quantitatively with UBER, leaving undetectable amounts of the native ODN, the excised ODN, and the truncated ODN on MALDI analysis. To evaluate multi-labeling efficiency, BETr labeling was performed with a dsDNA (4dU) containing four uracils (Supplementary Fig. 5). The results showed that all four uracils were replaced with CCVJ-1 in the ODN, leaving negligible amounts of under-labeled ODN, thus documenting sufficiently high efficiency for multi-labeling. A simple cost analysis of the process suggests that it is highly cost-efficient compared to a standard labeling approach (>100-fold less costly for single labeling in a 25-nt ODN, Supplementary Fig. 6), due to the low cost of enzyme and moderate cost of dU-containing ODNs as compared with chemically synthesized fluorescent ODNs.

Next, we tested a simultaneous in situ DNA synthesis and labeling process. For an ODN template containing nine adenine bases (9dA), a complementary primer was designed to bind at the 3′-end for initiating polymerase synthesis with deoxyuridine triphosphate (dUTP) replacing dTTP among the dNTPs[15]. Monitoring fluorescence intensity (538 nm) revealed a marked real-time fluorescence increase as the three steps proceeded simultaneously (Fig. 1d). Controls established that all three steps are needed for successful in situ synthesis and labeling. Moreover, this in situ DNA synthesis and BETr labeling was also applicable on RNA templates using reverse transcriptase (RTase) and specific primers (Fig. 1e). We know of no prior method that achieves DNA synthesis and covalent labeling in situ with light-up signaling on both DNA and RNA templates. Polymerase screening results showed that a lack of 3′ to 5′-exonuclease activity is beneficial for this labeling process, as is avoidance of high monovalent salt concentrations that interfere with UDG (Supplementary Fig. 7)[16]. The UBER-labeled ODN exhibited brighter fluorescence than achieved with Fl-dUTP (a common fluorophore for the labeling of ODNs) (Supplementary Fig. 8a–c).

We further envisioned the potential application of in situ synthesis and BETr labeling method to dsDNA obtained from cells, which can be useful for the preparation of FISH probes and blotting techniques[17, 18]. Genomic DNA (gDNA) extracted from HEK293T cells was subjected to nick translation and in situ BETr labeling (Fig. 1f). During the nick translation process, DNase I produces single-stranded nicks and polymerase I elongates the 3′ hydroxyl terminus, incorporating dU into the target and removing nucleotides by 5′ to 3′ exonuclease activity, then UBER labels the incorporated dU sites through BETr. We observed that all the reactions proceeded simultaneously in a single tube (Fig. 1g), and yielded a real-time fluorescence increase, reporting on reaction progress. The use of 100% dUTP in place of dTTP afforded the highest fluorescence enhancement, implying that all the sites opposite dA in gDNA were labeled with UBER (Fig. 1h). This labeling density is far higher than can be achieved with standard nick translation (Supplementary Fig. 8d). Moreover, control experiments with no dUTP showed negligible fluorescence enhancement, establishing that the levels of endogenous AP sites and dU in gDNA are low enough to yield very little background signal.

### Mechanistic study of in situ DNA synthesis and BETr labeling

We examined the effects of the relative numbers and positioning of labeling sites in DNA regarding the extent of fluorescence enhancement and labeling efficiency, employing template DNAs with varied numbers of adenines. In a 60-mer template context, 2–11 adenines were introduced, then subjected to the combined polymerase/BETr procedure (Fig. 2a, Supplementary Fig. 9). While more adenines generally resulted in higher fluorescence enhancement up to as many as nine adenines (9dA) among 42 bases, the sequence with 11 adenines (11dA) showed a slightly diminished fluorescence intensity. Considering that the labeling sites reside at every three or four nucleotides in 11dA (Supplementary Table 2), there were three possible reasons for the diminished fluorescence intensity in the ODN with more fluorophores: (i) inhibition of UDG by a nearby label, (ii) a self-quenching effect between adjacent dyes, and (iii) destabilization of the duplex by the fluorophores introduced at high density. First, possible inhibition

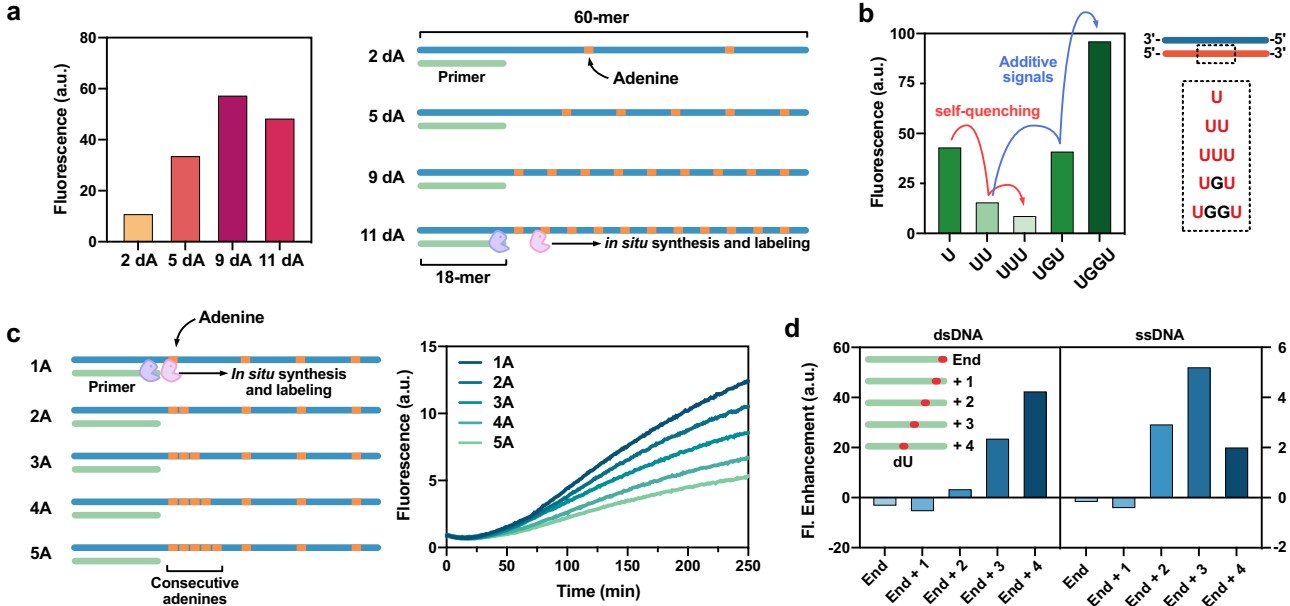

**Fig. 2 | Optimization of labeling frequency. a** Fluorescence intensity depending on the number of labeling sites in ODNs after 1000 min of reaction. **b** Fluorescence intensities of different numbers of CCVJ-1 in varied distances. **c** Fluorescence enhancement during the in situ synthesis and BETr labeling with the template ODNs with consecutive dA. **d** Fluorescence enhancement measured with DNAs containing one dU at varied positions from the end of the strand. [CCVJ-1] = 20 μM, [UDG] = 10 U/mL, [Templates] = 1 μM, [Primers] = 5 μM, [dATP, dCTP, dGTP] = 50 μM, [dUTP] = 100 μM, [Polymerase] = 2 U/mL. Data acquisition after 2.5 h incubation, measured in a Fluoroskan Ascent microplate reader (485/538 nm).

of UDG by a nearby label was tested using an ODN containing three consecutive dU nucleotides (Supplementary Fig. 10). The MALDI analysis results showed that all three consecutive sites were labeled with CCVJ-1/UDG without leaving any detectable under-labeled ODN, confirming no inhibitory effect of UDG by a nearby label already installed. To evaluate the quenching effect of the dye, dsDNAs with 1–3 consecutive dU or two dU at varied distances were prepared (Fig. 2b) and subjected to labeling. When fluorophores were introduced consecutively, the higher number of fluorophores exhibited lower fluorescence intensities, implying self-quenching at close distances (red arrow in Fig. 2b). Testing the effect of distance between two dye sites revealed that separation by two intervening base pairs gave little or no self-quenching (blue arrow in Fig. 2b). For closer substitutions, the introduction of UBER (which is larger than canonical bases) causes local destabilization of the duplex (Supplementary Fig. 11), which may place the dye in a partially single-stranded context where fluorescence intensity is reduced somewhat relative to fully doubled-stranded DNA[9]. Interestingly, this destabilization effect can be harnessed in turnover labeling (see below). Overall, we conclude that high UBER density and brightness can be achieved with a spacing of 2–3 nucleotides between sites, where the labeled DNA appears to be stable and self-quenching is minimal.

We evaluated the applicability of in situ DNA synthesis and BETr labeling for yet more dense fluorescence labeling on ODNs (Fig. 2c), testing ODNs encoding 1–5 consecutive labeling sites next to the primer binding site (1A–5A). We found that consecutive adenines in a template do not prevent success in labeling with the polymerase/BETr procedure. However, the intensity slightly decreases with added dye numbers, likely due to self-quenching and local DNA destabilization as noted above. We also tested BETr-labeling efficiency at the end of DNA strands, as the number of flanking base pairs alongside a lesion significantly affects the activity of glycosylases (Fig. 2d)[19, 20]. Labeling with UDG and CCVJ-1 gave fluorescence enhancement when lesions were located more than 2 nt away from the end, implying UDG requires at least two flanking bases alongside the lesion.

## Site-specific two-color labeling

Most DNA repair glycosylases are specialized to specific forms of damage, and recognize only a small number of related base lesions[21]. This fact raises the interesting possibility of a strategy for multiple orthogonal DNA repair reactions in one DNA for selective multi-color labeling, by use of two different lesions and two different repair glycosylases. As a test of this possibility, we developed an additional aminooxy-substituted rotor dye (UBER Red) (see details in the Supporting file, Supplementary Fig. 12). This dye also reacts with an AP site, affording 15-fold fluorescence enhancement at a red-shifted wavelength (ca. 620 nm) distinct from that of CCVJ-1 (Fig. 3a). For two-color labeling, we prepared a hairpin ODN (dU/dI) with two types of deaminated DNA lesions in the stem: (i) dU (a substrate of UDG), and (ii) deoxyinosine (dI) which is a substrate of N-methylpurine DNA glycosylase (MPG) (Fig. 3b). Adding MPG to the hairpin ODN successfully labeled it with UBER Green (CCVJ-1), and control experiments with a hairpin structure containing only dU (dU_sample) confirmed the dI excision specificity (Fig. 3c). The labeling capacity of UBER Red through BETr was also confirmed with UDG, showing red fluorescence with a negligible overlap in the green channel. We then doubly labeled the ODN in a site-specifically by consecutive treatment with each glycosylase/dye combination (Fig. 3c). The fluorescence intensities and the colors of labeled ODN documented that each dye was incorporated exclusively in the designated positions in a site-specific manner (Fig. 3c). Thus, the data show that dual labeling can be achieved by the use of two deaminated bases and enzymes selective for each.

## Synthesis of labeled DNAs on a sequence-specific target nucleic acid

We envisioned that synthesis of fluorescent-labeled DNA selectively on a complementary target DNA or RNA might be applied to the detection of nucleic acids among mixtures as well as to in situ synthesis of FISH probes in biological context, while removing the need for washing steps (Fig. 3d). To test this, a mixture of five spectator DNAs (41–60 nt) was used along with a specific target DNA (60nt), then the polymerase/BETr in situ labeling was carried out with a target-specific primer

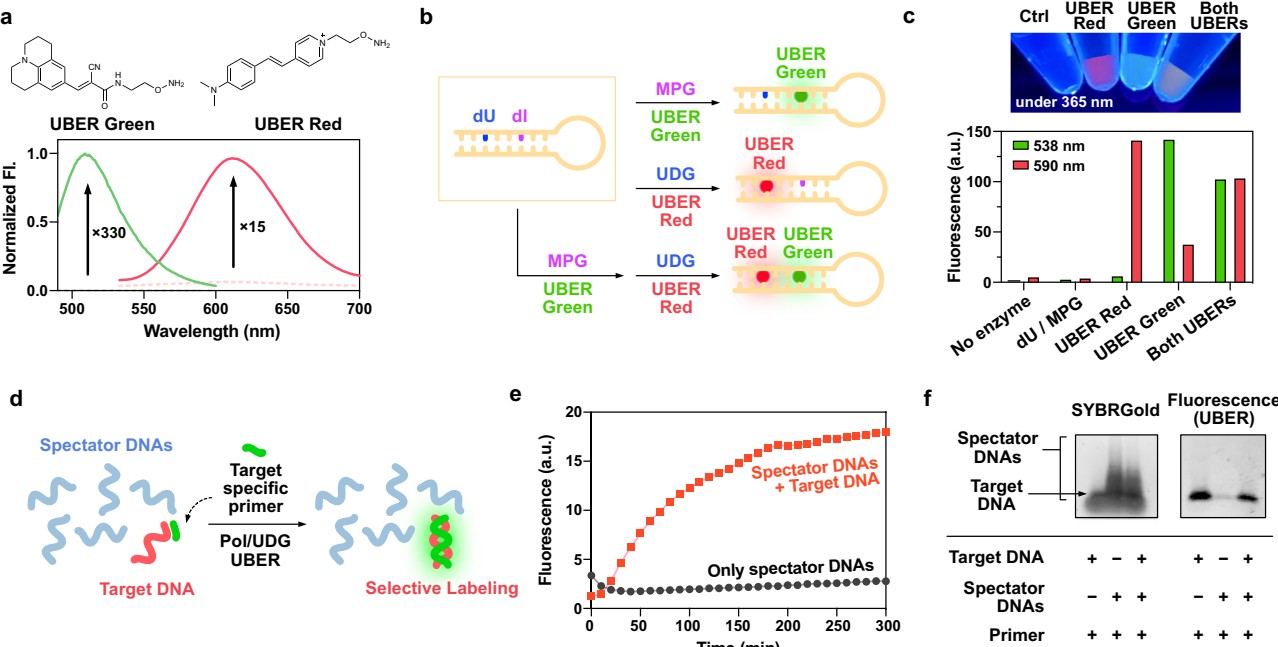

**Fig. 3 | Two-color labeling, and selectivity on a target DNA. a** Chemical structures and fluorescence spectrum of UBER Green and UBER Red showing enhancement in AP site. **b** Illustrational description of two-color labeling of a hairpin ODN containing two different DNA lesions with corresponding glycosylases. **c** A picture of UBER labeled ODNs purified by DNA precipitation taken on a gel illuminator with 365 nm excitation, and fluorescence intensities at 538 and 590 nm while excited at 485 nm. **d** Illustration of selective labeling of target DNA in the presence of spectator DNAs in the mix. **e** Fluorescence changes in the in situ synthesis and labeling of DNA in the presence of spectator DNAs with and without target DNA. **f** Agarose gel (1.5%) electrophoresis results showing selective labeling of the target DNA in the presence of spectator DNAs. [UBERs] = 10 μM, [hairpin oligo] = 1 μM, $\lambda_{ex}$ = 480 nm (UBER Green), 515 nm (UBER Red), [UDG] = 20 U/mL, [MPG] = 50 U/mL, [Target DNA] = 1 μM, [Spectator DNAs] = 1 μM each, [Target-specific primer] = 5 μM, [Klenow] = 2 U/mL, [dNTP (A,C,G)] = 50 μM, [dUTP] = 100 μM. The fluorescence spectra in **a** were measured on a Jobin Yvon-Spex Fluorolog 3. The fluorescence intensities were measured on **c**, **e** a Fluoroskan Ascent microplate reader (485 nm/538 nm) at 37 °C.

(Fig. 3e, Supplementary Table 2). Fluorescence analysis showed that the spectator DNAs lacking primer complementarity resulted in negligible covalent labeling, while the addition of a target DNA to the pool exhibited significant in situ fluorescence enhancement over time, documenting sequence-specific synthesis of fluorescently labeled DNA. Agarose gel analysis further confirmed that the in situ covalent labeling of UBER proceeded only in the presence of target DNA among the spectator sequences (Fig. 3f). High specificity was also obtained with an RNA template in the presence of background tRNA (Supplementary Fig. 13). This suggests the possible future utility of polymerase/BETr labeling for detection of targets in complex mixtures.

**Fluorescence signal reporting with isothermal amplification**
We next considered strategies involving the use of BETr for signaling isothermal amplification. Isothermal rolling circle amplification (RCA) rapidly generates long concatemeric copies of circular DNAs, and is widely employed in biomolecular target detection and amplification[22]. Given that a short DNA primer (10–15 nt) can be extended by a polymerase into a very long DNA strand (>12,000 nt)[23], we envisioned incorporation of many dU nucleotides with simultaneous BETr labeling, affording a long labeled DNA with high brightness. For this test, we employed M13mp18 circular ssDNA, ca. 7200 nt in length (Supplementary Fig. 14). Given the presence of consecutive adenine sites which would impede fluorescence enhancement in this circular DNA, we envisioned the use of mixtures of dUTP with dTTP to increase average spacing of labeling. When both dUTP and dTTP were provided in the reaction, the difference in ultimate fluorescence showed that dUTP and dTTP compete with each other, establishing that dUTP is close to dTTP in efficiency as a polymerase substrate (Fig. 4a). Interestingly, 50% dUTP in the mix resulted in ~75% fluorescence enhancement, suggesting highly favorable substrate behavior with this

enzyme. The results led us to test in situ RCA and BETr labeling with different ratio of dUTP to dTTP to optimize the ratio for efficient fluorescence labeling (Fig. 4b). In the beginning of the amplification (<100 min), a higher proportion of dUTP showed greater fluorescence enhancement, implying the faster incorporation and labeling. However, a low ratio (<2% dUTP) exhibited higher fluorescence enhancement at extended reaction times, plausibly due to the optimized frequency of fluorescence labeling (Supplementary Fig. 15).

**High-intensity multi-labeling of a DNA construct**
DNA nanostructures have had a remarkable impact on nano-science and -technology[24]. In this regard, a method that enables direct labeling of a DNA nanostructure after its 3D assembly holds significant potential, as dU replacing dT provides fully canonical base pairs, avoiding interference with hybridization and DNA assembly. Taking advantage of the multi-labeling and spacing information generated above, we proceeded to label a compact DNA construct into a bright fluorescence tag. Multi-labeling is broadly useful for sensitive analyte detection and imaging. Due to the utility of multiple labels, DNAs have been studied as scaffolds to host multiple dyes; an example includes the use of dsDNA stained with DNA-intercalating dyes[25], and self-assembly of multiple strands labeled with a single fluorophore at the end of the strands[26]. To test the labeling capability of BETr toward DNA assembly, tetrahedral DNA (TD) nanostructure was tested as a candidate for tagging. TDs are structurally compact and stable, are formed via simple self-assembly, and exhibit resistance to nuclease degradation[27]. A TD containing one deoxyuridine per 17 bp edge (six in total) was assembled from four ssDNAs, one of which was extended beyond the structure as a site for tethering via hybridization (Fig. 4c). BETr labeling of the assembled TD resulted in much greater fluorescence intensity than each ssDNA component (Fig. 4c), confirming the fluorescent

 

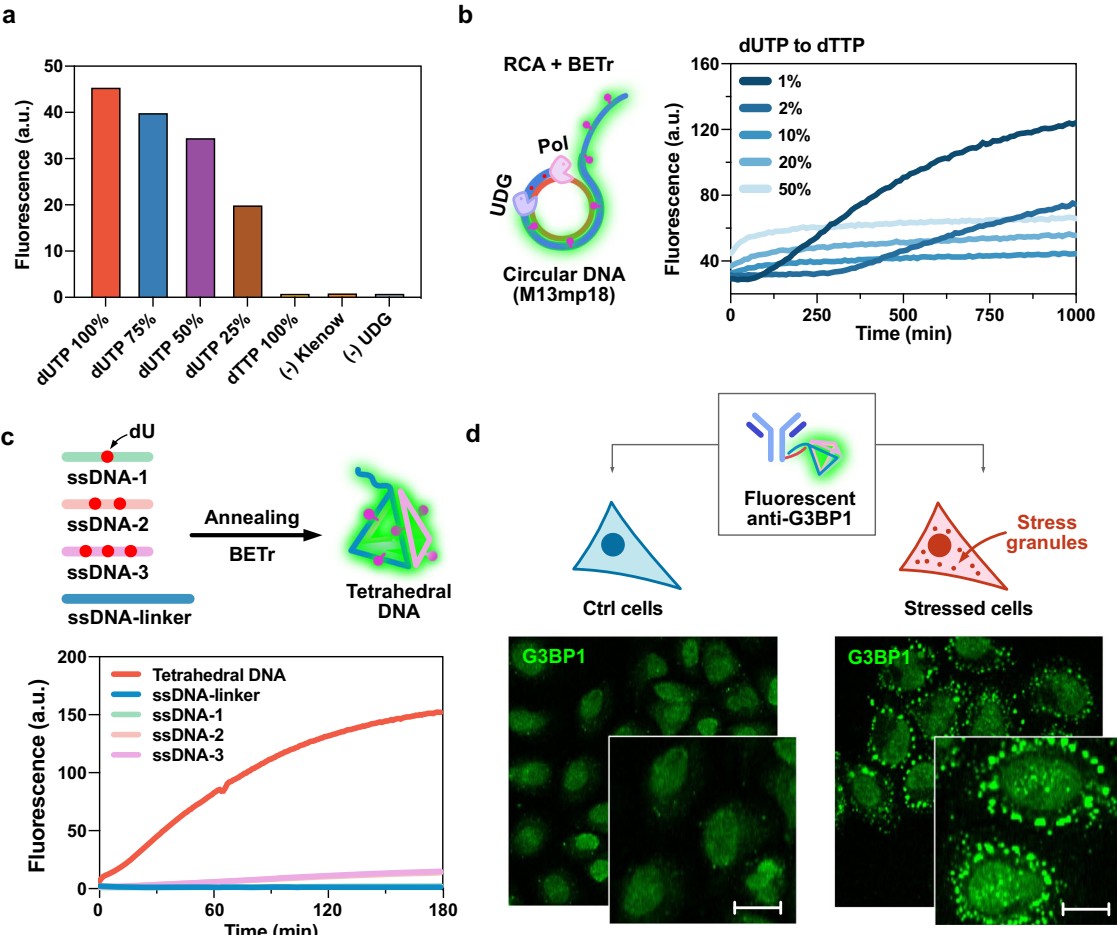

**Fig. 4 | Isothermal signal amplification and a bright DNA construct for tagging.** Strategies for intense labeling via multiple incorporation of UBER dye into DNAs. **a** Fluorescence intensity changes of in situ DNA synthesis and BETr labeling with different ratio of dUTP and dTTP. [9dA] = 1 μM, [Primer] = 5 μM, [dUTP] = 100 μM. **b** Fluorescence changes in RCA with M13mp18 and in situ BETr labeling depending on the relative ratio between dTTP and dUTP. **c** Fluorescence intensities in real-time BETr labeling of ssDNA components versus the assembled TD. **d** Direct immunofluorescence images of HeLa cells (control and stressed) with anti-G3BP1 tethered with the labeled tetrahedral DNA, visualizing stressed granules. [M13mp18] = 2.5 μg/mL, [RCA Primer] = 10 μM, [dATP, dCTP, dGTP] = 1 mM, [dUTP +dTTP] = 1 mM, [Polymerase] = 10 U/mL, [UDG] = 20 U/mL, [CCVJ-1] = 50 μM (**b**), 20 μM (**a**, **c**). [ssDNAs] = 2 μM. Scale bar = 20 μm. Bar was plotted with the data at 5 h of incubation. The fluorescence intensities were measured on a Fluoroskan Ascent microplate reader (485 nm/538 nm) at 37 °C.

advantage of a rigid duplex context for the UBER dye and labeling capability of BETr toward DNA assembly (Supplementary Fig. 16). Compared with fluorescein, a common and highly bright dye class[28], the CCVJ-1 labeled TD showed three-fold greater fluorescence intensity than a fluorescein-labeled oligonucleotide (Supplementary Fig. 17). It is likely that yet greater brightness could be achieved with constructs containing greater numbers of CCVJ-1 dyes, while multiple labeling of DNA with fluorescein can result in strong self-quenching (Supplementary Fig. 8)[8].

As a test of fluorescence tagging and imaging with the TD construct, we monitored the formation of stress granules in HeLa cells with a G3BP1 primary antibody conjugated with an oligonucleotide[29]. Stress granules are phase-separated aggregations in the cytosol composed of mRNAs and RNA-binding proteins[30], and the protein G3BP1 is employed commonly as a target to image the granules[31]. The stress granules were induced by the treatment of sodium arsenite for 30 min, and directly visualized with the TD labeled primary antibody for G3BP1 protein (Fig. 4d) by hybridization to the antibody's conjugated oligonucleotide. This enabled clear visualization of stress granules in cells without a secondary antibody, which can reduce the potential cross-reactivity in multianalyte imaging.

## Turnover detection of target DNA

We found that selective properties of repair enzymes can be taken advantage of in labeling strategies. For example, MPG, a bacterial enzyme that excises alkylated and deaminated purines, recognizes DNA lesions only in dsDNA, not in ssDNA[32]. Based on this dsDNA specificity, we designed a DNA detection system with the potential for turnover (Fig. 5a). In this detection system, a DNA probe containing dI, a substrate of MPG, in the center of the strand is designed to hybridize with a target DNA. In the absence of target DNA, the probe itself is a very poor substrate for MPG as it is single-stranded, yielding no fluorescence enhancement (Fig. 5b). On the other hand, in the presence of complementary target DNA, it forms an enzyme-competent duplex, and then is repaired and labeled by the MPG/UBER combination through BETr, affording strong fluorescence enhancement over time. Additionally, we tested varied bases opposite dI to optimize the activity of MPG (Supplementary Fig. 18), revealing the highest fluorescence enhancement when dI was paired opposite dT or dG, consistent with the reported kinetics of MPG[32]. Moreover, the detection system can also be applied to RNA detection (Supplementary Fig. 19). We found that the MPG enzyme also functions to excise the base from dI in DNA hybridized to RNA, enabling the detection of the target RNA sequence.

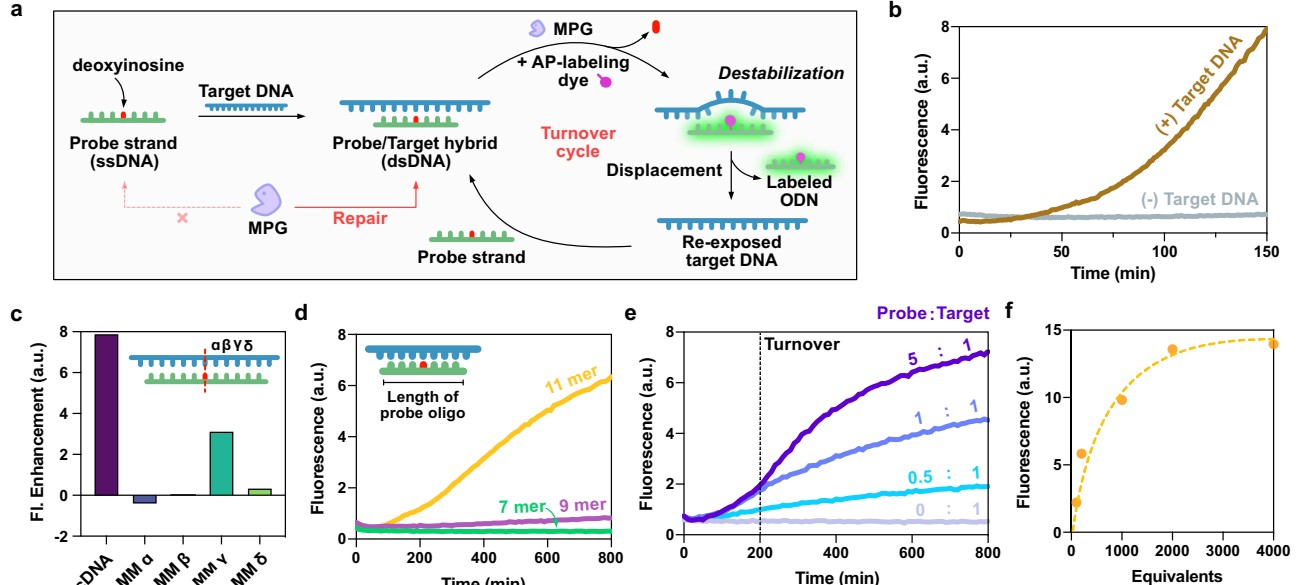

**Fig. 5 | Detection of a target through BETr with isothermal turnover and single nucleotide specificity. a** Illustration describing the mechanism of turnover detection strategy. **b** Fluorescence responses of labeling probe oligo (11mer) in the absence and presence of target DNA (Template dT). **c** Fluorescence enhancement upon the detection of target DNAs containing single mismatch in varied positions. **d** Fluorescence response depending on the lengths of probe oligos when detecting target DNA (Template dT). **e, f** Fluorescence response while detecting target DNA (Template dT) with 11mer depending on the relative amount of probe oligo and target DNA. [Target DNA] = 2 μM in **b–e**, 10 nM in **f**. [Probe oligo] = 10 μM (**b–d**), 200 nM–80 μM (**e–f**), [UBER] = 10 μM in **b–e**, 100 μM in **f**, [MPG] = 100 U/mL. The fluorescence intensities were measured on a Fluoroskan Ascent microplate reader (485 nm/538 nm) at 37 °C.

We further tested the single-mismatch discrimination of this strategy by preparing mismatched targets with mismatch positions varied 1–4 nt from the labeling site. Single mismatches 1 nt or 2 nt from the labeling site resulted in no fluorescence enhancement, thus showing very high specificity (Fig. 5c). Thus, the turnover detection system can be used to discriminate single nucleotide polymorphisms within this range. We hypothesize that the discrimination arises from the mismatch destabilization of the helix, preventing the formation of a stable duplex substrate for MPG. Mismatch stabilities depend both on the position as well as the specific bases involved, both of which were varied here. We note that the "gamma" site, which displayed less discrimination, was originally an AT pair, mutated to a relatively stable AG mismatch, while the others were GC pairs, likely resulting in larger $T_m$ differences.

Given that the dye-labeled position destabilizes the duplex, a DNA probe having optimized length may be long enough to form a semi-stable duplex that can act as a substrate for MPG, yet short enough to dissociate after the dye labeling destabilizes the duplex (Fig. 5a). We designed 7–11mer dI-containing probe strands and tested them for target DNA detection over time (Fig. 5d). The results revealed that at 37 °C a 7-mer strand was too short to act as a substrate, while an 11-mer exhibited robust repair activity with MPG and significantly greater fluorescence enhancement. To evaluate an application of the turnover design, fluorescence enhancement in this detection system was observed with different ratios of probe strand (11mer) to target DNA (Fig. 5e). The results showed that one copy of target DNA can label more than one copy of probe ODN. Moreover, an excess amount of probe ODN showed identical initial kinetics with an equivalent amount of probe ODN, indicating that only hybridized dsDNA was subject to BETr labeling and that the labeled single-stranded probe ODNs can turn over (Supplementary Fig. 17b). Additionally, we also investigated the detection of a sequence longer than 11 nt by introducing multiple deoxyinosines in the probe strand (Supplementary Fig. 20). The results showed that introducing multiple dI nucleotides with four base pairs between the lesions facilitated isothermal turnover detection for a longer sequence.

Testing with a lower 10 nM target DNA concentration (0.5 pico-mole) showed that five units of MPG facilitates ca. 2000-fold turnover of probe ODN in this condition (Fig. 5f), yielding greatly amplified signals relative to copy number. An advantageous feature of this turnover system is that the probe strand is completely non-fluorescent and bio-compatible, which enables the use of relatively high concentrations to detect a target at much lower concentration. The detection limit, in this case, was 0.5 femtomole of target (10 pM) using 10 μM probe DNA (Supplementary Fig. 21).

## Discussion

We have described a broadly applicable post-synthetic DNA labeling strategy that makes use of DNA repair enzymes generating AP sites at DNA lesions introduced beforehand and AP site-reactive rotor dyes (UBERs) that light-up upon reaction. DNA lesions (here in the form of uracil or inosine) can be introduced either chemically via phosphor-amidite reagents, or enzymatically with a nucleoside triphosphate and polymerase, to mark labeling sites. Notably, all components (including CCVJ-1 and oligonucleotides containing dU or dI) are commercially available and are thus accessible to the broader research community. The labeling method is carried out in a single tube, and signals its own progress with increasing fluorescence. The overall cost of the BETr method compares quite favorably to the use of commercial covalent dyes in DNA (Supplementary Fig. 6). As numerous other DNA fluorescence labeling methods have been developed over decades, many options exist for distinct applications[1]. For in vitro studies that can be done with very small amounts of labeled oligonucleotide (i.e., less than a nanomole), other commercially available DNA labeling methods also can be good options for applications such as several of those shown here. However, where large quantities (e.g., <2 nmol) of labeled oligonucleotides are required such as MERFISH in cells[33], the cost-efficiency of the BETr approach is favorable. We also note that like many other labeling methods, BETr does require a precipitation or spin column step to remove enzyme and excess dye; however, these are inexpensive and can be carried out in less than an hour.

An advantageous feature of this strategy is that all three reactions required for the labeling (insertion of DNA lesions, DNA repair, and AP site trapping) are orthogonal to each other and can proceed simultaneously, which enables in situ DNA synthesis and fluorescence labeling in a single-tube reaction by simply mixing all the components (Supplementary Fig. 22). Notably, the progress of the reactions can be monitored in real-time by fluorescence increases due to the light-up properties of UBER dyes as they react with the DNA. In the case of dsDNA labeling, all the reactions required for labeling double-stranded gDNA through nick translation are performed simultaneously in a single tube, affording remarkably high labeling efficiency in gDNA, and enabling the high-occupancy incorporation of many dyes. Moreover, the methods obviate tedious washing steps for in situ assays, as the glycosylase excises the deaminated bases only after the incorporation into DNA, and the CCVJ-1 dye fluoresces up to 500-fold more brightly in the dsDNA product. Combining two colors of UBER dyes and two damage/glycosylase pairs facilitated site-specific two-color labeling in one DNA, by performing BETr with dU and dI consecutively. A current limitation of BETr labeling (as compared with the use of commercial fluorescent dyes) is that relatively few high-performing AP-reactive chromophores are available thus far. Therefore, further studies will be required to develop additional UBER dyes with chromophores that offer alternative wavelengths. In this regard, the development of AP-reactive fluorophores with large light-up response upon labeling, modest self-quenching, and little quenching by DNA bases, will be beneficial. An additional limitation is that the brightness of a single UBER label is less than that of the brightest commercial dyes, and thus for single labeling, other dyes offer greater sensitivity. To achieve equivalent brightness, multiple UBER dyes are required in this case.

Sequence-specific primer extension enabled target-specific in situ detection and labeling with this labeling method. The robustness of UDG, which we find can repair dU in DNA/RNA contexts as well as dsDNA, enables in situ reverse transcription and fluorescence signaling to detect either DNA or RNA, broadening the applicability of the strategy. We further took advantage of the dsDNA specificity of the MPG enzyme to devise a robust isothermal turnover DNA detection system, in which labeled ODN is spontaneously displaced from the target DNA, continuously generating fluorescence signals.

We have also demonstrated bright labeling in single DNA constructs, with potential utility in reporting on isothermal amplification and in imaging applications. Our experiments confirmed labeling and light-up signaling of RCA via dUTP incorporation with optimum dUTP/dTTP ratios. For imaging, a compact and bright DNA nanostructure (TD) labeled with six molecules of CCVJ-1 showed high brightness, and tethering the fluorescent TD on a primary antibody for G3BP1 was exploited to successfully visualize stress granules in cells, without the need for a secondary antibody.

## Methods

### Oligonucleotides
Custom oligonucleotides were purchased from Integrated DNA Technology with HPLC purification, unless otherwise state. M13mp18 ssDNA was purchased from NEB. END was purchased from Protein and Nucleic Acid facility at Stanford.

### Measurement of fluorescence intensity of CCVJ-1
Unless otherwise noted, fluorescence intensity of CCVJ-1 was measured with the excitation of 485 nm (1 min$^{-1}$), while the emission was collected at 538 nm at 37 °C in a 384-well microplate with 50 μL of total volume in each well (Fluoroskan) or in a 96-well plate with a 200 μL total volume (Bio-Tek).

### General procedure for in situ synthesis and BETr labeling
To a template oligo (1 μM) dissolved in buffer D (see Fig. S6), 5 μM primer, 50 μM nucleoside triphosphates (dA, dU, dG, dC), 10 μM

UBERs, and 20 U/mL UDG (New England Biolabs) were added. Upon the addition of 2 U/mL polymerase (Klenow), the reaction started at 37 °C and was observed on a plate reader (Fluoroskan). For experiments with RNA, a 20 mM tris buffer pH 8.0 with 4 mM MgCl$_2$ and 1 mM DTT was used.

### Procedure for agarose/PAGE gel analysis
After the reaction, 10 μL (or equal volume of reaction) loading dye (8 M Urea, 0.05% Orange G, 0.05% Bromophenol blue) was added and the mixture was loaded on a denaturing 12% agarose gel or 15% PAGE. DNAs were separated in an agarose gel or PAGE gel in 1× TBE buffer (pH 8.3), 15 W, ~2 h. The DNA was visualized by fluorescence imaging (Typhoon, GE Healthcare, or Blue light transilluminator).

### Procedure for nick translation and dsDNA BETr labeling
HEK293T cells purchased from ATCC were cultured in DMEM supplemented with 4 g/L D-glucose, L-glutamate, 100 mg/L sodium pyruvate, 10% fetal bovine serum (FBS), 0.1 U/L penicillin, and 0.1 U/L streptomycin in a humidified incubator at 37 °C with 5% CO$_2$. The cells were passaged when they reached ~80% of confluency. Then gDNA was extracted from the cells using Easy-DNA gDNA purification kit (Invitrogen). 2 μg of gDNA was subjected to nick translation using Nick translation DNA labeling system 2.0 (Enzo Life Sci.) according to the protocol provided except for the addition of dTTP as the concentration is not disclosed. In addition to this setup, 20 μM UBER, 20 U/ml UDG, and 100 μM of dUTP+dTTP were added for the nick translation and BETr labeling at 37 °C.

### Procedure for filtration using size exclusion centrifugal filter
After carrying out the general procedure for in situ DNA synthesis and fluorescence labeling, the reaction was loaded into a 3k centrifugal filter and the filter was filled with DI water. The mixture was filtered using a centrifuge (13,800 × g for 12 min) at 4 °C. The filter was washed and spun four times to completely remove the remaining dyes. At the fifth cycle, the same procedure was applied, spinning for 30 min to concentrate the purified oligonucleotide. Then the purified oligonucleotide was collected and the fluorescence intensity was measured on a Fluoroskan plate reader (485 nm/528 nm).

### Preparation of G3BP1 primary antibody conjugated with tethering oligonucleotide
Oligonucleotide for tethering (tethering oligo)[29] was purchased from Alpha Thera and 100 μg of primary antibody for G3BP1 was purchased Sigma-Aldrich (SAB4500043-100 μg). The oligonucleotide was reconstituted in ddH$_2$O, and 50 μL of the solution was added to 50 μg of antibody. After thoroughly vortexing it, the mixture was placed on the ice sideways, then irradiated with 365 nm hand-held UV light for 2 h.

### Preparation of TDN and tethering with oligo-labeled G3BP1 primary antibody
2 μM of ssDNA-1, ssDNA-2, ssDNA-3, and 2.2 μM ssDNA-linker was annealed and labeled with 20 μM of CCVJ-1 and 20 U/mL of UDG in buffer A (see Fig. S7) for 12 h at 37 °C. Labeled DNA was purified by size exclusion centrifugal filters (10 kDa), then mixed with the tethering oligo-labeled antibody.

### Cell experiments
HeLa human cervical cancer cells (from ATCC) were cultured in DMEM supplemented with 4 g/L D-glucose, L-glutamate, 100 mg/L sodium pyruvate, 10% FBS, 0.1 U/L penicillin, and 0.1 U/L streptomycin in a humidified incubator at 37 °C with 5% CO$_2$. The cells were passaged when they reached ~80% of confluency. The cells were seeded onto a FluoroDish, 35 mm tissue culture dishes with 10 mm cover glass bottom, at a density of 1.0 × 10$^6$ cells, and incubated overnight at 37 °C with 5% CO$_2$. The cell culture medium was removed, then the cells were

washed with PBS three times and fixed with 200 µL of 4% methanol-free formaldehyde for 10 min at room temperature. The formaldehyde was removed, and the cells were washed with PBS three times. Then, the cells were permeabilized with 0.5% Triton-X in PBS for 10 min at room temperature. After the medium was removed, the cells were washed with PBS, then incubated with 3% BSA in PBS for 1 h at room temperature. After the cells were washed with PBS, the cells were incubated with the TDN tethered primary antibody for G3BP1 in PBS (1:500 dilution) containing 1% BSA for 24 h at 4 °C. After the staining, the cells were washed with PBS three times, then mounted on the confocal microscope. A 25× oil immersive objective was employed for the imaging. TDN tethered antibody was visualized with a 488 nm laser with 500–615 nm. Images were processed and analyzed in Image-J software.

### Statistics and reproducibility

To establish the reproducibility of the BETr-labeling method, each labeling procedure was repeated at least three times and the results were almost identical to each other (see supplementary Fig. 2). For the target-specific labeling experiments, the fluorescence measurements were repeated twice with virtually identical results, and one of the samples was used for the gel analysis. For the visualization of stress granules in cells, varied fixation and labeling procedures that are commonly used for immunostaining were tested (data not shown), and the stress granules were clearly visualized with all procedures tested, confirming reproducible labeling of the primary antibody with the UBER labeled fluorescent DNA construct. The electrophoresis experiments provided in the supplementary information were repeated three times with almost identical results, and one typical result was presented in the article. Samples sizes are listed in the manuscript, and no samples were excluded. No data were excluded from the analyses.

### Reporting summary

Further information on research design is available in the Nature Research Reporting Summary linked to this article.

## Data availability

The raw data used in the manuscript is available in a public repository (https://doi.org/10.6084/m9.figshare.20421915.v1). Source data are provided with this paper.

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

## Acknowledgements
We acknowledge support from the U.S. National Cancer Institute (R01 CA217809). E.M.H. was supported by grants from the Le Moyne Research and Development Committee.

## Author contributions
Y.W.J. collected data and wrote the manuscript. E.M.H. performed the RNA-related experiments and contributed to the writing of the manuscript. L.X. performed the gel electrophoresis. D.L.W. contributed to the collection of data. E.T.K. led the project as P.I. and contributed to the writing of the manuscript.

## Competing interests
The authors declare no competing interests.
