## [Peer Review File · Nature Communications]

Reviewers' Comments:

Reviewer #1:

Remarks to the Author:

The manuscript by Yong Woong Jun et al., "Efficient DNA fluorescence labeling via base excision trapping" describes post-synthetic DNA labeling method, named by the authors base excision trapping (BETr), based on specific labeling of generated AP sites in duplex DNA with a small molecule fluorescence probe (UBER).

This probe was previously developed by the authors to assess the activity of DNA glycosylases in real time in vitro and in living cells. This topic is of broad scientific interest due to the development of new tools for cell imaging and specific nucleic acid detection. The data presented in the manuscript indicated that BETr labeling based on incorporation of deaminated bases (dU, dI) in DNA, their excision by DNA glycosylases and covalent modification of the resulting AP sites with a UBER probe can be efficiently performed in the same reaction mixture (in situ). Using DNA as a scaffold to accommodate multiple dyes, the authors synthesized a UBER-labeled tetrahedral DNA complex as a fluorescent label and after hybridizing it with an oligonucleotide-conjugated antibody demonstrated that such labeling can be used to visualize certain components in fixed cells without secondary antibodies. The use of BETr with isothermal turnover for specific detection of short nucleotide sequences in the target DNA seems also promising. Overall, the work is interesting and of high quality, but in my opinion, it does not provide significant progress in DNA labeling tools and the benefits of its use in practice are very unclear.

The main points to be addressed

1. All of the DNA targets used in the study are single-stranded oligonucleotides or circular M13mp18, easily serving as a template for primer binding and subsequent BETr. It is unclear how this works with natural DNA samples, which are double-stranded and may contain some deaminated bases or AP sites. The authors need to demonstrate the utility of BETr for labeling fixed cell chromatin and purified genomic DNA compared to other traditional labeling tools.
2. To prove the advantage of imaging with TD constructs, the authors should demonstrate example(s) where the use of this "high-intensity" tag was the only way to detect a low content target protein, instead of visualizing stress granules, which can easily be done with traditional approaches.
3. Similarly, high molecular weight dsDNA should be used to demonstrate the ability of the "BETr with isothermal turnover" approach to specifically detect short sequences. What is the detection limit of the tool?

Specific comments:

What was the concentration of "spectator DNAs" in the experiments in Figure 3?

The cost of TAMRA-labeled oligo (Supplementary Fig. 5) is overestimated (μmol instead of nmol ?). Moreover, the cost of the proposed TD structure containing multiple dU residues may end up being more expensive than an oligonucleotide with a conventional fluorescent dye.

Reviewer #2:

Remarks to the Author:

The manuscript describes labeling of apurine/apyrimidines (AP) sites of DNA/RNA with turn-on fluorescence by using simple, biocompatible and cheap starting materials in a single reaction tube. This strategy uses base excision repair enzymes to remove deaminated bases to generate AP sites. These sites react with UBER dyes to generate a strong fluorescent signal. This strategy is diversified in various streams of DNA/RNA labeling such as hairpin DNA, circDNA, tet.DNA etc and validated experimentally. These experiments are elegantly designed and narrowed down for a real application. This strategy has a real potential for practical implementation. However, the same strategy was already reported in their recent publications (doi.org/10.1002/anie.202111829, DOI:

10.1021/jacs.9b09812). These results are new direction to these recent reports.

In a sample already having a/few AP site's, without the need of enzymes/dU/dI, UBERs react with these sites and generate fluorescent signal. Determination of how much damage was there in an unknown sample is still a challenging question with these experiments. How do the authors explain this requirement?

The authors did not mention anywhere about experimental repetitions /reproducibility.

Page3: "This new methodology includes" this is not new, it's a strategy already reported in their recent publications (doi.org/10.1002/anie.202111829, DOI: 10.1021/jacs.9b09812).

Page3, Results: "enhancement upon binding with AP sites in DNA." UBERs are not binding to AP sites, but reacting with AP sites. Covalent binding could be appropriate word.

Page5: "dye, 40% of the excised DNA was fragmented," and thereof, how authors concluded as 40% (0.5%, 1.5% ..) fragmentation with mass spectrometric analysis? The authors mentioned using the heights of mass peaks (suppl.Fig.3) which is not a convincing method. LCMS could be relatively helpful.

Page6: "separation by two intervening base pairs" can authors comment why dG is preferred over other nucleobases?

Page6: In Fig.2 legend, dTTP+dUTP is used, authors did not discuss its importance in text. Do they use dTTP+dUTP in all experiments described in Fig.2? because number of AP sites would be differed and contradict the explanations in relevant Fig.2 main text.

Page8: "50% dUTP in the mix resulted in ~75% dUTP incorporation" How it is calculated?

Comparing the florescence with dUTP selectivity (Fig 4a) is not appropriate, since the fluorescence is not proportional to number of UBER incorporations. The authors already described that with increasing number of UBER labeling, the fluorescence is different/diminished.

Page10: "Thus, the turnover detection system" In Fig 4c, the authors did not comment on the gamma position which has 30-40% fluorescent signal compared to cDNA and these results are concluded for whole SNP. Additionally, the enzyme recognize the AP sites preferentially having deaminated bases, however, 5-methyl cytosine (5mC:G) will result in thymine (T G), which cannot be recognized by deaminase enzymes. In this context, how would the authors resolve this? This is also valid for other deaminase products such as hypoxanthine, xanthine.

Page10: "homogeneous detection system" What exactly do the authors want to convey here? Please clarify in text.

Page10: " The results showed enhanced signal with more than one copy of labeled ODN relative.."

Difficult to understand what the authors are trying to convey. Please reframe or explain alternatively. What is meant by equivalents in Fig 5f? How can a copy number and 2000-fold turnover be explained from Fig 5f?

Page11: In Fig 5e, how did the authors determine the 1st round after 200 min? Assumptions in a text is fine, but in a dataset it is not good unless determined experimentally.

In Fig 4d, mention the magnification or indicate length of scale bar in cell image or in legend

Please mention time in bar plots, for example in Fig 4a and Supp.Fig 14a and all other places in manuscript, the plots are at what time point? Since the florescence intensity is changing with time. Supplementary Table 1. List of oligonucleotides. Color labeling of sequences are missing. (for eg., dU are marked in orange, and primer binding sites are marked in green).

The authors used different fluorescence instruments (Jobin Yvon-Spex Fluorolog 3 spectrometer, Fluoroskan Ascent microplate fluorometer, Bio-Tek Synergy HT microplate fluorimeter). It would be really helpful for readers if the authors could provided in the supplementary information which data/fig are from which instrument. For example, can you comment on Y-axis of Fig 3e and Suppl.Fig.9? These are hugely differed by intensity.

Reviewer #1 _____

The manuscript by Yong Woong Jun et al., “Efficient DNA fluorescence labeling via base excision trapping” describes post-synthetic DNA labeling method, named by the authors base excision trapping (BETr), based on specific labeling of generated AP sites in duplex DNA with a small molecule fluorescence probe (UBER).

This probe was previously developed by the authors to assess the activity of DNA glycosylases in real time in vitro and in living cells. This topic is of broad scientific interest due to the development of new tools for cell imaging and specific nucleic acid detection. The data presented in the manuscript indicated that BETr labeling based on incorporation of deaminated bases (dU, dI) in DNA, their excision by DNA glycosylases and covalent modification of the resulting AP sites with a UBER probe can be efficiently performed in the same reaction mixture (in situ). Using DNA as a scaffold to accommodate multiple dyes, the authors synthesized a UBER-labeled tetrahedral DNA complex as a fluorescent label and after hybridizing it with an oligonucleotide-conjugated antibody demonstrated that such labeling can be used to visualize certain components in fixed cells without secondary antibodies. The use of BETr with isothermal turnover for specific detection of short nucleotide sequences in the target DNA seems also promising. Overall, the work is interesting and of high quality, but in my opinion, it does not provide significant progress in DNA labeling tools and the benefits of its use in practice are very unclear.

We thank the reviewer for the appreciation of our work and for the valuable comments. In response to the last comment questioning the benefits of this work, we have prepared a comparison table (now added to the supporting information (Supplementary Table 1)). Several features stand out for the new method, including low cost, light-up response, density of labeling, and turnover labeling. Also, please see the comment below on Q1.

Supplementary Table 1. Comparison of features of BETr labeling and common DNA labeling methods

	BETr	Phosphoramidite		Triphosphate		MTase	TGT	β-GT
		Fluorophore	Click handle	Fluorophore	Click handle			
Labeling long DNA	✓	✗	✗	✓	✓	✓	✓	✓
One-step labeling	✓	✓	✗	✓	✗	✓	✓	✓
In situ synthesis & labeling	✓	✓	✗	✓	✗	✗	✗	✗
Light-up	✓	✗	Δ (Rare)	✗	Δ (Rare)	✗	✗	✗
Turnover labeling	✓	✗	✗	✗	✗	✓	✗	✗
Cost efficiency	High	Expensive modified phosphoramidites		Expensive modified triphosphates		Very expensive modified cofactors		
Enzyme compatibility	High	n/a	n/a	Low	High	High	High	High
Reagent availability	Commercially available					Synthesis required for cofactors		
Labeling site specificity	Δ only with synthetic template	High		Low	Δ only with synthetic template	High	High	High
Consecutive Multi-labeling	✓	✗	✓	✗	✓	✗	✗	✗
Requirement for synthetic chemistry lab	No	Yes	Yes	No	Yes	No	No	No

Q1. All of the DNA targets used in the study are single-stranded oligonucleotides or circular M13mp18, easily serving as a template for primer binding and subsequent BETr. It is unclear how this works with natural DNA samples, which are double-stranded and may contain some deaminated bases or AP sites. The authors need to demonstrate the utility of BETr for labeling fixed cell chromatin and purified genomic DNA compared to other traditional labeling tools.

We believe that our writing emphasizing “post-synthetic labeling of DNA” may have led to unclear conceptions regarding the end-goal of this work. There are two main applications in the field of covalent fluorescence labeling of DNA: (i) preparation of customized fluorescence-labeled oligonucleotides, and (ii) covalent labeling of target biological DNAs. Chemical synthesis with modified phosphoramidites and enzymatic synthesis with modified nucleoside triphosphates have been mainly utilized for (i), while methyltransferase and glucosyltransferases with modified cofactors have been used for (ii).¹ The significant advantage of this work, which is an optimized procedure for (i), is being able to synthesize and fluorescent label custom oligonucleotides simultaneously without any chemistry setup by simply mixing all the components, with high speed and cost efficiency (Supplementary Table 1). We revised the main text to clarify the goals and the practical use of this work.

Having said that, we also appreciate the reviewer’s comment suggesting the labeling of naturally-derived dsDNA. Prompted by this, we have now applied the new approach to label genomic DNA through nick translation. gDNA was extracted from HEK293T cells and labeled via nick translation in the presence of BETr labeling components (Figure R1a). We observed that (1) creating nicks with DNase I, (2) replacing nucleotides (incorporation of dU) by polymerase I, (3) excising dU by UDG, and (4) trapping the AP sites by UBER all proceed simultaneously in a single tube (Figure R1b). Indeed, the approach shows two benefits over standard labeling by nick translation: it can be carried out with much higher density of labeling (we used up to 100% dUTP, whereas commercial kits use 1% fluorescent dUTP due to the bulkiness of the modified dUTP and resistance of polymerase to it), and the reaction occurs with a real-time light-up signal, documenting labeling progress. The new results have been added to the main text.

Figure R1. *In situ* nick translation and BETr labeling of duplex genomic DNA. (a) Illustration of *in situ* nick translation and BETr labeling in a single reaction. (b) Real-time fluorescence enhancement of UBER during *in situ* nick translation and BETr labeling depending on the relative concentrations of dUTP and dTTP. [UBER] = 20 μM, [dUTP+dTTP] = 100 μM, [UDG] = 20 U/mL, and incubation at 37 °C. The fluorescence was measured on a Fluoroskan Ascent microplate reader (485 nm/538 nm)

Importantly, the new results showed that the oligonucleotide labeled with BETr showed almost three-fold higher fluorescence intensity than the one with FI-dUTP. To further evaluate the labeling density with FI-dUTP, we tested *in situ* synthesis of duplex DNA and labeling with an oligonucleotide with five consecutive adenines (5A) (Figure R2d). When 100% FI-dUTP was used without dilution with dTTP, PAGE gel electrophoresis data showed that DNA synthesis stalled at the consecutive incorporation sites, almost certainly due to the bulkiness of incorporated fluorescein. Note that BETr labeling method labeled consecutive labeling sites with remarkably high yield (Supplementary Fig. 9, Fig. 2c). Diluting the concentration of FI-dUTP to 20% with dTTP somewhat improved polymerase progression; however, a significant portion of the DNA remained stalled in the consecutive-site region. Given that natural gDNA contains many consecutive adenine sites, BETr offers higher labeling density than virtually any other method.

Figure R2. Comparison of BETr labeling of *in situ*-synthesized duplex DNA with conventional labeling method by FI-dUTP. (a) Illustration of *in situ* synthesis and labeling with BETr and FI-dUTP, and following washing step for the measurement of brightness. (b) Normalized fluorescence changes while FI-dUTP or UBER are incorporated into newly synthesized oligonucleotides. (c) Fluorescence intensities of the oligonucleotides labeled with BETr or FI-dUTP, after the filtration step removing remaining unreacted dyes. (d) 15% Denaturing PAGE gel electrophoresis after the DNA synthesis with 5A and primer under fluorescence imaging setup with FITC channel. The fluorescence was measured on a Fluoroskan Ascent microplate reader (485 nm/538 nm). [9dA] = 1 μ M, [5A] = 1 μ M, [Primer] = 5 μ M, dNTP(A,C,G) = 100 μ M, [dUTP] = 100 μ M, [FI-dUTP] = 10 μ M, [UBER] = 10 μ M, UDG = 20 U/ml, [Klenow] = 2 U/mL.

The reviewer's comment regarding possible interference by endogenous DNA damage and AP sites in natural DNA is reasonable. They are known to emerge and undergo repair continuously in cells. However, the endogenous level of deoxyuridine/AP sites is extremely low compared to the BETr labeling density employed here. Control experiments with 100% dTTP, which also included UDG in the reaction mixture, showed an almost identical fluorescence profile as UBER probe itself without gDNA (Figure R1b), documenting that endogenous AP sites and uracil in gDNA do not contribute significant background interference under these conditions.

Q2. To prove the advantage of imaging with TD constructs, the authors should demonstrate example(s) where the use of this "high-intensity" tag was the only way to detect a low content target protein, instead of visualizing stress granules, which can easily be done with traditional approaches.

The reviewer is correct that secondary antibodies can be used to image stress granules. The new TD fluorescence tag-based immunostaining illustrates the advantage of direct visualization compared to the use of a secondary antibody, and visualization of stress granules with the fluorescent tag was a demonstration of versatility of this method. “High-intensity” in the text was intended to mean a bright tagging strategy for labeling a DNA construct compared to traditional single-fluorophore labeling, and can be applied to the DNA nanotechnology field as well as immunostaining. The main text has been revised to clarify the utility (as compared with the use of secondary antibodies) and versatility of a labeled TD construct beyond immunostaining.

Q3. Similarly, high molecular weight dsDNA should be used to demonstrate the ability of the “BETr with isothermal turnover” approach to specifically detect short sequences. What is the detection limit of the tool?

This detection system based on isothermal turnover has been developed for the detection of single-stranded nucleic acids (DNA/RNA), not double-stranded ones. In principle, the single-stranded specificity of this detection/labeling system can be useful for the detection of single-stranded regions in RNAs, ssDNA viruses, or ssDNA exposure during DNA replication. Moreover, this detection system is not necessarily for the detection of short sequences only, as it can also be applied, in principle, for the detection of longer sequences by adding multiple labeling sites. Prompted by the reviewer’s comment, we designed longer DNA probes with two deoxyinosines with different gaps between the two lesions (Figure R3a). The results showed that longer DNA probes with multiple deoxyinosines with 4 bases between the inosines can be utilized for the isothermal turnover system, while placing deoxyinosines further than 5 bases from each other showed no turnover capacity (Figure R3b).

Figure R3. Isothermal turnover detection with extended DNA probes containing two deoxyinosines at different positions. (a) Illustration of longer probe DNA design with multiple dI. (b) Fluorescence enhancement under the BETr labeling condition with 1 equivalent of target DNA (3bp, 4bp, 5bp) in the presence of 1 or 5 equivalents of probe DNA strand. [Long Target DNA] = 2 μ M, [Probe DNA] = 2 or 10 μ M, [UBER] = 20 μ M, [MPG] = 50 U/mL. The fluorescence intensities were measured on a Fluoroskan Ascent microplate reader (485 nm/538 nm) after 5 h of incubation.

Also in new experiments, we tested compatibility of the detection system for reporting on single-stranded RNA sequences via transient DNA/RNA hybrids. To the best of our best knowledge, the base excision capability of *N*-methylpurine DNA glycosylase (MPG) toward deoxyinosine in DNA hybridized with RNA has never been explored. In this test, we found that MPG can excise the deoxyinosine in a DNA/RNA hybrid and subsequently UBER can label the new AP site, which documents that this detection system also can be applied for RNA detection (Figure R4).

Figure R4. Normalized fluorescence enhancement of isothermal turnover detection with target RNA and probe DNA (11 mer). [Target RNA] = 10 μ M, [Probe DNA] = 10 μ M, [UBER] = 10 μ M, [MPG] = 50 U/mL. The fluorescence intensities were measured on a Fluoroskan Ascent microplate reader (485 nm/538 nm) at 37 °C.

Addressing the reviewer’s comment about detection limit, we measured the limit of detection of a target DNA (**Template dT**). When tested with 10 μ M Probe DNA (**11mer**), 100 U/mL MPG (New England Biolab.), and 100 μ M UBER, the LOD was observed to be 10 pM (0.5 femtomole) in a 50 μ L volume (Figure R5). This has also been added to the manuscript.

Figure R5. Detection limit of the isothermal turnover detection system. [Template dT] = 1 pM–1 nM, [11mer] = 10 μ M, [UBER] = 10 μ M, [MPG] = 100 U/mL. The fluorescence intensities were measured on a Fluoroskan Ascent microplate reader (485 nm/538 nm) at 37 °C.

Q4. What was the concentration of “spectator DNAs” in the experiments in Figure 3?

Concentrations of all spectator DNAs and target DNA were 1 μ M each. The information has been added to the legend of figure 3.

Q5. The cost of TAMRA-labeled oligo (Supplementary Fig. 5) is overestimated (μ mol instead of nmol ?). Moreover, the cost of the proposed TD structure containing multiple dU residues may end up being more expensive than an oligonucleotide with a conventional fluorescent dye.

We were also surprised by the fact that fluorophore addition cost was much higher than that of deoxyuridine. For the comparison, we have attached here an actual quote from IDT which has a reputation for reasonable pricing in the US. From the company, a 25mer oligonucleotide with canonical bases costs a base price of \$17.50 (guaranteed

amount is 35 nmol). When substituting with deoxyuridine, it costs extra \$13.30 for the same guaranteed amount, while adding TAMRA at the same position costs extra \$201.30 for much less guaranteed amount (2 nmol) and even requires additional HPLC purification (+\$40.00) plausibly due to the low reaction yield (Figure R6).

Figure R6. Quotes for purchase of synthetic oligonucleotides showing the relative costs of including deoxyuridine and TAMRA.

Similarly, for a longer oligonucleotide (55mer used for TD construction) costs \$44.00 with canonical bases (28 nmol of guaranteed amount), and addition of two deoxyuridines to the oligo costs an extra \$26.40 (\$13.20 each) for the same guaranteed amount, while addition of one TAMRA costs extra \$231.20 (+\$44.00 for HPLC purification) with only 1.5 nmol of guaranteed amount, which is much less. In summary, addition of six deoxyuridine in total costs less than \$80.00 for 28 nmol, while the addition of single TAMRA dye costs more than \$200.00 for 1.5 nmol of guaranteed amount, which further documents the cost efficiency of BETr labeling method (Figure R7).

Figure R7. Quote for purchase of synthetic oligonucleotides showing the relative commercial costs of adding deoxyuridine and TAMRA from IDT.

Reviewer #2 _____

The manuscript describes labeling of apurine/apyrimidines (AP) sites of DNA/RNA with turn-on fluorescence by using simple, biocompatible and cheap starting materials in a single reaction tube. This strategy uses base excision repair enzymes to remove deaminated bases to generate AP sites. These sites react with UBER dyes to generate a strong fluorescent signal. This strategy is diversified in various streams of DNA/RNA labeling such as hairpin DNA, circDNA, tet.DNA etc and validated experimentally. These experiments are elegantly designed and narrowed down for a real application. This strategy has a real potential for practical implementation. However, the same strategy was already reported in their recent publications (doi.org/10.1002/anie.202111829, DOI: 10.1021/jacs.9b09812). These results are new direction to these recent reports.

We thank the reviewer for the appreciation of our work and for the valuable comments.

Q1. In a sample already having a/few AP site's, without the need of enzymes/dU/dI, UBERs react with these sites and generate fluorescent signal. Determination of how much damage was there in an unknown sample is still a challenging question with these experiments. How do the authors explain this requirement?

This is a fair concern. As the reviewer pointed out, for a random DNA sample containing AP sites, UBER will label with the AP sites and generate fluorescence signal (it is noteworthy that this methodology was developed to readily synthesize/prepare customized fluorescence-labeled oligonucleotides, rather than labeling unknown samples). However, the chances that a random sample contains significant amount of AP sites are slim as endogenous depurination generating AP sites is a relatively slow and rare occurrence, and is continuously repaired in cells. In control experiments for labeling gDNA through nick translation (Figure R1b, see Q1 of Reviewer 1), gDNA extracted from HEK293T containing endogenous levels of AP sites and dU showed negligible labeling efficiency in the presence of UDG and UBER compared to the labeling through *in situ* dU incorporation and BETr labeling. In an extreme case where high levels of AP sites or DNA damage was intentionally introduced in a sample, pre-treatment of alkoxyamine compounds could likely be used to mask the pre-exist AP sites. Alternatively, a commercially available DNA repair mix might also be used to bypass the issue.

Q2. The authors did not mention anywhere about experimental repetitions /reproducibility.

Reproducibility is very high; we have now tested this directly and commented on it in the text. Triplicate BETr labeling of an oligonucleotide (10 μ M, dU-sample), UBER (10 μ M), and UDG (20 U/mL) showed almost identical fluorescence trajectories (Figure R8a). In addition, a plot testing varied concentrations of the oligonucleotide shows a very good line fit with little scatter (Figure R8b,c).

Figure R8. Reproducibility of BETr labeling. (a) Fluorescence enhancement of 10 μ M dU-sample incubated with 20 U/mL UDG and 10 μ M UBER at 37 $^{\circ}$ C. (b) Fluorescence enhancement of dU-sample at different concentration when incubated with 20 U/mL UDG and 10 μ M UBER at 37 $^{\circ}$ C. (c) Fluorescence intensity depending on the concentration of dU-sample after 60 min of incubation. The fluorescence intensities were measured on a Fluoroskan Ascent microplate reader (485 nm/538 nm) at 37 $^{\circ}$ C.

Q3. Page3: “This new methodology includes” this is not new, it’s a strategy already reported in their recent publications (doi.org/10.1002/anie.202111829, DOI: 10.1021/jacs.9b09812).

The sentence is now advised accordingly.

Q4. Page3, Results: “enhancement upon binding with AP sites in DNA.” UBERs are not binding to AP sites, but reacting with AP sites. Covalent binding could be appropriate word.

We have revised the statement as the reviewer suggested.

Q5. Page5: “dye, 40% of the excised DNA was fragmented,” and thereof, how authors concluded as 40% (0.5%, 1.5% ..) fragmentation with mass spectrometric analysis? The authors mentioned using the heights of mass peaks (suppl.Fig.3) which is not a convincing method. LCMS could be relatively helpful.

As the reviewer suggests, quantification of oligonucleotides with MALDI-TOF can be biased due to the existence of non-homogeneity of sample distribution within a spot or large discrepancy in flight probability resulting from high molecular weight. We tried our best to bypass those issues by averaging more than 20 scans from each spot and by not using oligonucleotide heavier than 11000 Da which usually begin to show low flight probability. To further confirm that 40% of oligonucleotides were truncated after the excision, we have now analyzed the formation of AP site and its truncation using PAGE gel analysis (Figure R9). Gel analysis results also showed that the excision by UDG was almost quantitative without remaining substrate and 40–50% of the oligonucleotide with AP site was truncated. In addition, we removed the quantification part measured by MALDI-TOF from the main text to reduce the concern.

Figure R9. Denaturing PAGE gel electrophoresis analysis of dU-containing oligonucleotide (dU-sample) before and after incubating with 20 U/mL UDG for 24 h. The gel was visualized with SYBR gold.

Q6. Page6: “separation by two intervening base pairs” can authors comment why dG is preferred over other nucleobases?

We have not chosen dG over other nucleobases for a specific reason; the counter bases were coincidentally dC. We believe that the intervening bases could be any bases. Our prior work in DNA repair analysis showed that excision and labeling occur with virtually any adjacent base.

Q7. Page6: In Fig.2 legend, dTTP+dUTP is used, authors did not discuss its importance in text. Do they use dTTP+dUTP in all experiments described in Fig.2? because number of AP sites would be differed and contradict the explanations in relevant Fig.2 main text.

We appreciate the comment. In the Figure 2 experiments, only dUTP was used. The legend is now edited accordingly.

Q8. Page8: “50% dUTP in the mix resulted in ~75% dUTP incorporation” How it is calculated? Comparing the fluorescence with dUTP selectivity (Fig 4a) is not appropriate, since the fluorescence is not proportional to number of UBER incorporations. The authors already described that with increasing number of UBER labeling, the fluorescence is different/diminished.

The point is well taken. We have only compared the fluorescence enhancement, and our claim might not be appropriate as the reviewer pointed out. We have revised the sentence accordingly.

Q9. Page10: “Thus, the turnover detection system” In Fig 4c, the authors did not comment on the gamma position which has 30-40% fluorescent signal compared to cDNA and these results are concluded for whole SNP.

The mechanism of a single mismatch discrimination in this experiment is expected to be the destabilization effect of the duplex so that the probe strand becomes a less effective substrate for MPG. We believe that the positional variation in selectivity arises from the T_m differences in mismatches as compared with original base pairs; mismatch stabilities depend both on position as well as the specific bases involved, both of which were varied here. Note that

the “gamma” site was originally an AT pair, mutated to a relatively stable AG mismatch, while the others were GC pairs, likely resulting in larger T_m differences. A comment has been added to the main text to point this out explicitly.

Q10. Additionally, the enzyme recognize the AP sites preferentially having deaminated bases, however, 5-methyl cytosine (5mC:G) will result in thymine (T G), which cannot be recognized by deaminase enzymes. In this context, how would the authors resolve this? This is also valid for other deaminase products such as hypoxanthine, xanthine.

In this work, we used already-deaminated bases with glycosylases to excise the lesions, and do not employ deaminase enzymes. BETr labeling can be employed with any damaged base that can be excised by glycosylase, but not with all kinds of deaminated bases. For example, we have shown that hypoxanthine in DNA (deoxyinosine) can be labeled through BETr using MPG (Figure R10), and our previous report showed that AP sites can be also generated with dU/SMUG1, 8-oxo-dG/OGG1, and 5-OH-dU/NTH1.⁴ As far as we know, there is no reported glycosylase for 5mC, but other methylation variants such as 5-formyl-cytosine (fdC) can be also used for BETr labeling with TDG enzyme (Figure R10). This opens future opportunities for multiplexing.

Figure R10. BETr labeling of oligonucleotides with dU or 5-formyl-dC (fdC). [Oligonucleotides] = 1 μ M, [TDG] = 38 nM, [UBER] = 10 μ M. The fluorescence intensities were measured on a Fluoroskan Ascent microplate reader (485 nm/538 nm) at 37 °C.

Q11. Page10: “homogeneous detection system” What exactly do the authors want to convey here? Please clarify in text.

The sentence has been revised for clarity.

Q12. Page10: “ The results showed enhanced signal with more than one copy of labeled ODN relative..” Difficult to understand what the authors are trying to convey. Please reframe or explain alternatively.

The sentence has been revised for better readability.

Q13. What is meant by equivalents in Fig 5f? How can a copy number and 2000-fold turnover be explained from Fig 5f?

The “equivalents” in Fig. 5f referred to the equivalents of probe ODN relative to target ODN. Given that MPG only repairs DNA lesions in dsDNA, MPG/UBER exhibiting light-up signal with up to 2000 equiv. of probe ODN in the presence of 1 equiv. of target ODN implies 2000-fold turnover of target ODN (Note that the turnover here is not referring the turnover of enzyme but the target ODN). We have revised the sentence to minimize the misunderstanding regarding what turnover represents in this section.

Q14. Page11: In Fig 5e, how did the authors determine the 1st round after 200 min? Assumptions in a text is fine, but in a dataset it is not good unless determined experimentally.

We agree. The figure has been revised accordingly.

Q15. In Fig 4d, mention the magnification or indicate length of scale bar in cell image or in legend Please mention time in bar plots, for example in Fig 4a and Supp.Fig 14a and all other places in manuscript, the plots are at what time point? Since the florescence intensity is changing with time.

Supplementary Table 1. List of oligonucleotides. Color labeling of sequences are missing. (for eg., dU are marked in orange, and primer binding sites are marked in green).

The authors used different fluorescence instruments (Jobin Yvon-Spex Fluorolog 3 spectrometer, Fluoroskan Ascent microplate fluorometer, Bio-Tek Synergy HT microplate fluorimeter). It would be really helpful for readers if the authors could provide in the supplementary information which data/fig are from which instrument. For example, can you comment on Y-axis of Fig 3e and Suppl.Fig.9? These are hugely differed by intensity.

We appreciate the comment. All the missing information is now added to where it is supposed to be. Note that fluorescence intensity was still measured with arbitrary units. Therefore, even if two different plots were measured on the same instrument, the values in different plots are not directly comparable.

References

- 1 Klöcker, N., Weissenboeck, F. P. & Rentmeister, A. Covalent Labeling of Nucleic Acids. *Chem. Soc. Rev.* **49**, 8749–8773 (2020).
- 2 Malicka, J., Gryczynski, I. & Lakowicz, J. R. Enhanced Emission of Highly Labeled DNA Oligomers Near Silver Metallic Surfaces. *Anal. Chem.* **75**, 4408–4414 (2003).
- 3 Lindahl, T. & Nyberg, B. Rate of Depurination of Native Deoxyribonucleic Acid. *Biochemistry* **11**, 3610–3618 (1972).
- 4 Wilson, D. L. & Kool, E. T. Ultrafast Oxime Formation Enables Efficient Fluorescence Light-Up Measurement of DNA Base Excision. *J. Am. Chem. Soc.* **141**, 19379–19388 (2019).

Reviewers' Comments:

Reviewer #1:

Remarks to the Author:

The revised manuscript has been greatly improved by the addition of some new experiments as well as improved clarity of presentation. However, it should be taken into account that:

- 1) the strategy itself is not new, as already mentioned by reviewer 2 (notably, the authors still leave in the discussion " We have described a novel and broadly applicable post-synthetic DNA labeling strategy...");
- 2) Brightness/sensitivity advantages are still limited (there are other fluorescent dyes that are much brighter than the fluorescein that was taken for comparison);
- 3) the cost difference may only be important if a relatively large amount of labeled oligonucleotide is required for the experiment, in most cases 1 nmol is sufficient. Notably, the synthetic oligonucleotides are homogeneous, pure, and single-stranded, ready-to-use, unlike the BETr reaction mixture;
- 4) the authors claim that this is a widely useful tool, but they have not demonstrated its use in an example problem that would be harder (or more laborious, or significantly more expensive) to solve with other known methods.

This is why I think this manuscript deserves publication, but in a more specialized journal than Nature Communications.

Reviewer #2:

Remarks to the Author:

This manuscript describes fluorescent labelling of abasic (AP) sites of DNA and RNA for practical application in relatively more efficient manner compared to the existing fluorescent labelling tools. In this method, authors introduce deaminated bases (dUTP, and dITP) during polymerase synthesis of target DNA/RNA and create AP sites using excision repair enzymes (UDG, MPG). These AP sites are trapped with universal base excision reporters (UBER) and produce AP specific fluorescent labelled DNA or RNA with minimal background signal and without washing. This is best performed with some requirements such as 2-3 nucleotide spacing between the UBER substrates and duplex stability. However, these can be overcome by using mixture of (dUTP+dTTP) during DNA synthesis. Cell viability experiments to trace localization of AP-sites in presence of DNA repair enzymes in living cells was reported in their recent publication(s) (doi.org/10.1002/anie.202111829, [doi/10.1021/jacs.9b09812](https://doi.org/10.1021/jacs.9b09812)). These progresses could potentially influence the field of labelling nucleic acid specifically AP sites.

In the revised manuscript, the authors pointed out the comments and modified the manuscript with new relevant experiments from previous submission. In new experiments, the labelling of genomic DNA (gDNA) using BETr labelling tool further prove the orthogonal compatibility. With the new experiments and revisions, the manuscript is suitable for publication after minor revisions.

Suppl. Fig. 4. In addition to Suppl. Fig. 4, authors should perform PAGE analysis having dU-sample+UDG+UBER. This will further provide a strong support that UBER link is more stabilizing and no/less truncated. Please remove the repeating words from figure legend.

Provide full Gel image of Fig.3f in SI.

Cite relevant references for Nick translation, FISH probes and blotting techniques. (Nick translation is widely applicable to DNA relevant transcriptions, it is interesting how the term translation is used in nick translation).

Provide the structure of FI-dUTP (SEEBRIGHT Green 496 dUTP) to indicate the position of modification in reference to Supp. Fig. 8d.

Label the y-axis in Suppl. Fig. 9.

Reviewer #1

The revised manuscript has been greatly improved by the addition of some new experiments as well as improved clarity of presentation.

We appreciate the reviewer's suggestions, which led to the improvement of the manuscript. We understand that the reviewer would like more cautions about the limitations of the labeling method; this has now been done at multiple places in the text and supporting file.

Q1. the strategy itself is not new, as already mentioned by reviewer 2 (notably, the authors still leave in the discussion " We have described a novel and broadly applicable post-synthetic DNA labeling strategy...")

The main text now has been revised accordingly to remove the word "novel" in the discussion.

Q2. Brightness/sensitivity advantages are still limited (there are other fluorescent dyes that are much brighter than the fluorescein that was taken for comparison)

Point well taken; when it comes to single dyes, it is true that there exist other dyes brighter than fluorescein. We have now added a note that brighter dyes do exist and may yield different comparative results.

We are not claiming that UBER (of which core fluorophore is CCVJ1) is the brightest dye. We are simply pointing out that the very high density of labeling that can be achieved and low self-quenching properties of the CCVJ-1 dye can yield total fluorescence (in multi-labeled DNA) that is higher than some other approaches. We chose fluorescein for comparison because CCVJ-1 uses the same filter sets, and fluorescein remains a popular dye.

We have added an additional caution to the Discussion section, noting the color limitations of UBER (there are many colors of commercial fluorescent dyes available for DNA, but only a small number thus far for UBER labeling).

Q3. the cost difference may only be important if a relatively large amount of labeled oligonucleotide is required for the experiment, in most cases 1 nmol is sufficient. Notably, the synthetic oligonucleotides are homogeneous, pure, and single-stranded, ready-to-use, unlike the BETr reaction mixture

It appears that the reviewer is referring to short labeled oligonucleotides here. We agree that the importance of cost difference will vary depending on the scale of the experiments. We have now included a caution that the cost comparison will vary based on scale and supplier. Note that we have included two commercially available scales in our analysis, and our costs were normalized per nanomole. (Note that one cannot order just 1 nmol of labeled oligonucleotide to our knowledge).

We have also added a note about purification in the supporting file, and have also added an explicit note about it in the Discussion section as well. For other applications of UBER labeling (such as in enzymatic DNA synthesis/labeling), purification would be needed for both approaches to remove excess dye and enzyme.

The reviewer may have missed the fact that UBER labeling can be done on single-stranded oligonucleotides (see Fig. 4c).

Q4. the authors claim that this is a widely useful tool, but they have not demonstrated its use in an example problem that would be harder (or more laborious, or significantly more expensive) to solve with other known methods.

We understand the concern. Respectfully, we believe that the labeling (done here) that would be harder to achieve with other known methods include (i) *in situ* enzymatic DNA synthesis and labeling while monitoring the process with light-up signal, (ii) target-sequence specific *in situ* DNA synthesis and labeling, (iii) isothermal turnover fluorescence labeling selectively in the presence of target DNA/RNA, (iv) very high efficiency of consecutive labeling, (v) high-yield multi-labeling of long oligonucleotides (>200 nt), and (vi) nick translation labeling of gDNA with 100% replacement at every “T” position. We believe that these experiments with other methods would be more laborious or more expensive (for cost analysis see Fig. S6).

Reviewer #2

This manuscript describes fluorescent labelling of abasic (AP) sites of DNA and RNA for practical application in relatively more efficient manner compared to the existing fluorescent labelling tools. In this method, authors introduce deaminated bases (dUTP, and dITP) during polymerase synthesis of target DNA/RNA and create AP sites using excision repair enzymes (UDG, MPG). These AP sites are trapped with universal base excision reporters (UBER) and produce AP specific fluorescent labelled DNA or RNA with minimal background signal and without washing. This is best performed with some requirements such as 2-3 nucleotide spacing between the UBER substrates and duplex stability. However, these can be overcome by using mixture of (dUTP+dTTP) during DNA synthesis. Cell viability experiments to trace localization of AP-sites in presence of DNA repair enzymes in living cells was reported in their recent publication(s) (doi.org/10.1002/anie.202111829, [doi/10.1021/jacs.9b09812](https://doi.org/10.1021/jacs.9b09812)). These progresses could potentially influence the field of labelling nucleic acid specifically AP sites.

In the revised manuscript, the authors pointed out the comments and modified the manuscript with new relevant experiments from previous submission. In new experiments, the labelling of genomic DNA (gDNA) using BETr labelling tool further prove the orthogonal compatibility. With the new experiments and revisions, the manuscript is suitable for publication after minor revisions.

We appreciate the reviewer's suggestions, which led to the improvement of the manuscript.

Q1. Suppl. Fig. 4. In addition to Suppl. Fig. 4, authors should perform PAGE analysis having dU-sample+UDG+UBER. This will further provide a strong support that UBER link is more stabilizing and no/less truncated. Please remove the repeating words from figure legend.

We appreciate the suggestion, and we now have done this. The stabilization effect on AP sites via the trapping by UBER was analyzed by PAGE gel. When 10 μM dU-sample was labeled with 5 μM of UBER, both AP site and truncated sample started to decrease with the appearance of UBER labeled oligo (Supplementary Fig. 4b, lane 3). After labeling all 10 μM dU-sample by using 20 μM of UBER, there remained only negligible amounts of the AP site and truncated one (Supplementary Fig. 4b, lane 4). The new data are now available in revised Supplementary figure 4. The legend is revised as well.

Supplementary Figure 4. Denaturing PAGE gel electrophoresis analysis of dU-containing oligonucleotide, showing stabilization of AP site in the presence of UBER reagent. (a) dU-sample (10 μM) was incubated with 20 U/mL UDG for 24 h. The gel was visualized with SYBR gold. (b) dU-sample (10 μM) was incubated with 20 U/mL UDG in the presence of UBER (5 μM lane 3, and 20 μM lane 4) for 24 h. The gel was visualized under a gel illuminator (465 nm).

Q2. Provide full Gel image of Fig.3f in SI.

The full gel image of Fig.3f has been added to Supporting Information Fig. S11, as suggested.

Supplementary Figure 11. Full gel image of target specific BETr labeling Agarose gel (1.5%) electrophoresis results showing selective labeling of the target DNA in the presence of spectator DNAs. [UBERs] = 10 μM , [UDG] = 20 U/mL, [Target DNA] = 1 μM , [Spectator DNAs] = 1 μM each, [Target specific primer] = 5 μM , [Klenow] = 2 U/mL, [dNTP (A,C,G)] = 50 μM , [dUTP] = 100 μM .

Q3. Cite relevant references for Nick translation, FISH probes and blotting techniques. (Nick translation is widely applicable to DNA relevant transcriptions, it is interesting how the term translation is used in nick translation).

The related references are now cited in the main text.

Q4. Provide the structure of Fl-dUTP (SEEBRIGHT Green 496 dUTP) to indicate the position of modification in reference to Supp. Fig. 8d.

The chemical structure of Fl-dUTP has been added to Supplementary Figure 8c.

Q5. Label the y-axis in Suppl. Fig. 9.

The y-axes in Fig. S9 have been labeled.